# Prediction of Walking and Arm Recovery after Stroke: A Critical Review

**DOI:** 10.3390/brainsci6040053

**Published:** 2016-11-02

**Authors:** Li Khim Kwah, Robert D. Herbert

**Affiliations:** 1Discipline of Physiotherapy, Graduate School of Health, University of Technology Sydney, Ultimo, NSW 2007, Australia; 2Neuroscience Research Australia (NeuRA), Randwick, NSW 2031, Australia; r.herbert@neura.edu.au

**Keywords:** prediction, prognosis, recovery, walking, arm function, stroke

## Abstract

Clinicians often base their predictions of walking and arm recovery on multiple predictors. Multivariate prediction models may assist clinicians to make accurate predictions. Several reviews have been published on the prediction of motor recovery after stroke, but none have critically appraised development and validation studies of models for predicting walking and arm recovery. In this review, we highlight some common methodological limitations of models that have been developed and validated. Notable models include the proportional recovery model and the PREP algorithm. We also identify five other models based on clinical predictors that might be ready for further validation. It has been suggested that neurophysiological and neuroimaging data may be used to predict arm recovery. Current evidence suggests, but does not show conclusively, that the addition of neurophysiological and neuroimaging data to models containing clinical predictors yields clinically important increases in predictive accuracy.

## 1. Introduction

It would be useful to be able to predict recovery of walking and arm after stroke. Accurate predictions are needed so that clinicians can provide patients with prognoses, set goals, select therapies and plan discharge [1,2,3,4]. For example, if it was possible to predict with some certainty that a particular patient would be unable to walk independently at six months, the clinicians providing that patient with acute and subacute care might work toward a discharge goal of safe transfers. Therapy might involve carer training and equipment prescription rather than intensive gait training. The ability to make accurate predictions could reduce the length of stay in hospitals and enable efficient utilization of stroke care resources [4,5].

Several systematic reviews have identified strong predictors of walking and arm recovery after stroke [2,3,6]. In one systematic review of prognostic studies on walking, clinical variables such as age, severity of paresis and leg power were found to be strong predictors of walking after stroke (based on five studies, each of between 197 and 804 patients) [2]. In another systematic review of prognostic studies on arm recovery, clinical, neurophysiological and neuroimaging data were found to be strong predictors of arm recovery after stroke (based on 58 studies of 9–1197 patients) [3]. These clinical, neurophysiological and neuroimaging data included measures of upper limb impairment, upper limb function, lower limb impairment, motor and somatosensory evoked potentials, and measures obtained with diffusion tensor imaging [3].

In practice, clinicians base their predictions about clinical outcomes on multiple variables [7,8,9]. If multiple predictors are to be used to make prognoses, there needs to be a proper accounting of the independent (incremental) predictive value of each predictor variable. Therefore the most useful information about prognosis is likely to come from multivariate prediction models [7,8,9].

The research which underpins establishment of clinically useful multivariate prediction models involves several steps. First ‘development studies’ are conducted to build the multivariate prediction models [7]. Subsequently the predictive accuracy of the models is tested on new cohorts [7,10]. These studies are known as ‘validation studies’ [7]. It is recommended that prediction models should not be used in clinical practice until both development and validation studies have been conducted [7,10]. Once development and validation studies have been conducted, impact studies may be conducted, although the reality is that few reports of impact studies are published. Impact studies resemble clinical trials; they test the efficacy of use of prediction models on patient outcomes, clinician behaviour and cost-effectiveness of care [7,11]. Recent narrative reviews have provided updates on the prediction of motor recovery after stroke [5,12] but these reviews have not focused on development and validation studies of models for predicting walking and arm recovery.

This review provides a critical review of prediction models of walking and arm recovery after stroke. Studies were identified using the search strategy and inclusion criteria in the Appendix A. The review begins in the second section with the definitions and measurements of walking and arm recovery. The third section provides a detailed description of the recommended process for developing and validating a prediction model because this process provides a benchmark against which prediction modelling studies of walking and arm recovery can be evaluated. The fourth section critically appraises development and validation studies of walking and arm recovery with the aim of identifying multivariate models that could potentially be implemented in clinical practice. Much has been written about the role of neurophysiological and neuroimaging data in predicting arm recovery. The fifth section considers whether neurophysiological and neuroimaging data provide additional predictive value over clinical data alone in predicting arm recovery. We conclude with a summary and recommendations for future prediction modelling studies. 

## 2. Definitions and Measurements of Walking and Arm Recovery

Most studies investigating recovery of walking after stroke have focused on the ability to walk independently [13,14,15,16,17,18,19,20,21,22]. The operationalization of this construct varies slightly with regard to the distance covered, the type of aids needed and the type of surfaces. For example, independent walking has been defined as “independence of gait and stair climbing with or without the use of some form of assistive device” [16], “walks 5 m without any assistance or walks with aid” [22] and “imperfect gait, but could be independently ambulant, usually with a walking aid” [15]. Various measurement tools have been used to measure walking in stroke. Some authors have used walking-specific measurement tools like the Functional Ambulation Category [14,16,17,20,23], the Rivermead Mobility Index [19,24] or the walking items of scales that measure stroke severity (e.g., Scandinavian Stroke Scale) [22], motor function (e.g., Motor Assessment Scale) [25] or activities of daily living (e.g., the Barthel Index or the Functional Independence Measure) [21,26,27,28,29,30,31,32,33]. Despite the heterogeneity of these measurement tools, it is easy to extract data on walking from them. Independent walking is equivalent to a score of ≥4/5 on the Functional Ambulation Category [14,16,17,20,23], a score of ≥3/6 on item 5 of the Motor Assessment Scale [25], a score of ≥9/12 on the gait item of Scandinavian Stroke Scale [22] and a score of 15/15 on the mobility item of the Barthel Index [28,29]. Studies that measured walking outcomes were included in this review.

For arm recovery, there is less consensus on an operational definition. It is generally agreed that measures of arm recovery should reflect a patient’s ability to use his or her arm in everyday life [6,34,35,36]. This is certainly considered an important goal to patients after stroke [37]. It is difficult, however, to define arm recovery due to the wide range of actions or tasks associated with functional use of the arm. Therefore the term arm recovery in this review will be used broadly to include measures of body functions or structure (e.g., muscle strength) or activities (e.g., ability to carry a tray of food across a crowded room) [6,38]. Various measurement tools used in the literature to measure arm recovery include arm-specific measurement tools like the Action Research Arm Test [39,40,41,42,43], the Nine Hole Peg Test [43,44,45,46] and the upper limb items of scales that measure body functions or structures (e.g., Motricity Index [45], Fugl-Meyer [45,47]) or activities (e.g., Motor Assessment Scale [25,30,48]). In prognostic studies in stroke, arm recovery has been defined by cut-off scores or change scores on these scales. Examples include cut-off scores of 10/56 [41,42], 35/56 [39], 50/56 [49] on the Action Research Arm Test, 5/6 on upper limb items 7 and 8 of Motor Assessment Scale [25], 18/18 on upper limb items 6 to 8 of Motor Assessment Scale [48] and change scores of upper limb items of Motricity Index [45] and Fugl-Meyer [45,47]. The problem with attributing a score to the definition of arm recovery is that readers have to be familiar with the scale to know what the score represents. Some have argued that a low score (e.g., 10/56 on the Action Research Arm Test) may not be reflective of meaningful arm function [5,39]. Some authors have also used activities of daily living scales (e.g., the feeding, dressing and toileting items of the Barthel Index) to measure arm recovery [50,51]. These scales are, however, insensitive to recovery of function in the hemiparetic arm since they allow use of the intact arm [6,34,52]. For this reason, tests that allow use of both arms (e.g., the Wolf Motor Function Test, the Frenchay Arm Test, the Arm Motor Ability Test and the Bell Test) may be less preferred as measures of arm recovery [52]. Studies that measured outcomes of hemiparetic arm recovery were included in this review. Studies that used activities of daily living scales or tests that allow use of both arms were not included in this review.

## 3. How Should Prediction Models Be Developed and Validated?

In recent years, several excellent guidelines have been published to guide the conduct and reporting of prognostic studies [53,54,55] and prediction modelling studies [8,9,56]. These guidelines make recommendations about the processes of model development and validation. Adherence to the processes recommended in these guidelines should reduce bias in predictions and maximise the applicability of the models to clinical practice. For model development, we focus on three elements recommended in all guidelines. They are: (a) recruitment of representative cohorts; (b) use of clearly defined, standardised and easily accessible predictors; and (c) careful selection of predictors for inclusion in the model. For model validation, we focus on two key elements: (a) the need for external validation; and (b) the reporting of predictive accuracy in a way that allows for comparison of the performance of models across cohorts in both development and validation studies.

To obtain representative cohorts it is necessary to sample an inception cohort consisting of consecutive cases. This means that all (or nearly all) eligible participants must be recruited at a uniform time point in the course of the disease [57,58]. Non-consecutive sampling in cohort studies can result in selective sampling (so that, for example, only patients with atypically good or atypically bad prognoses are included in the study) which potentially leads to the development of models that cannot be applied to the wider population of interest [55,56]. The second element to consider when developing a model is the use of clearly defined and standardised predictors. Heterogeneity in the definition and measurement of predictors potentially reduces the accuracy of predictions and may cause otherwise potentially useful predictors to be excluded from the final prediction mode [56]. It is also important to ensure that predictors are easily accessible and available at the time the model is to be used, as this enhances the generalisability and applicability of models to clinical practice [7]. The third element to consider is the careful selection of predictors when building a multivariate model. While there are many variable selection procedures, and none is considered universally superior to others [59] it is clear that the use of simplistic variable selection procedures (e.g., forward selection techniques) can lead to predictor selection bias [56], particularly if the sample size is small or the number of potential predictors is large [60,61], if continuous predictors are dichotomised [62,63], or if there is no accounting in the selection process for the number of alternative models considered by the variable selection algorithm [56,59]. To reduce the risk of overfitting, contemporary variable selection techniques such as shrinkage techniques and bootstrapping are recommended to adjust the weights assigned to the selected predictors [56,64,65]. These procedures should increase the accuracy of predictions made when the model is applied to new cohorts [56,64,65].

Validation of a development model involves testing the model on a new group of patients, preferably patients with similar characteristics located in a different hospital or centre [10]. This is called external validation. External validation is more rigorous than internal validation (using data splitting techniques or bootstrapping techniques to determine the sensitivity of model parameters to randomness in the sample) or temporal validation (applying the model to new patients in same hospital or centre) [10]. If comparisons are to be made between the development cohort and the validation cohort, the calibration and discrimination of the models must be quantified [56]. Calibration refers to how well observed probabilities agree with predicted probabilities. These probabilities are often plotted against each other so that, with perfect predictions (i.e., observed probabilities matching predicted probabilities), the observations fall along a line [65]. Discrimination refers to how well the model can distinguish between patients with good and bad outcomes and is often quantified with the concordance (c) statistic or the area under the receiver operating characteristic (ROC) curve [65]. Other measures of model performance such as the Hosmer-Lemeshow test (for calibration) and sensitivity and specificity (for discrimination) are widely used but are less useful for quantifying model performance [56,65].

## 4. Critical Appraisal of Development and Validation Studies of Walking and Arm Recovery

### 4.1. Walking and Arm Recovery: Critical Appraisal of Development Studies

It is important that models are developed on representative acute stroke cohorts. The cohorts may be obtained from registries [13,22,29] or may consist of consecutive patients admitted to hospitals with stroke [25,44,45,66,67,68]. However a large proportion of prognostic studies/models on walking and arm recovery have been developed on patients included in clinical trials [16,41,46,69,70,71,72,73] or referred for rehabilitation [14,17,18,19,23,24,26,27,28,33,40,51,74,75,76,77,78,79,80,81,82,83]. This is a problem because clinical trials often select participants using strict inclusion criteria that do not render the sample representative of the wider stroke population, and rehabilitation cohorts may have prognoses that are not typical of all patients. Use of these samples potentially biases predictive models, or at least limits the predictions to populations like those sampled in these studies. Models developed on trial cohorts or rehabilitation cohorts might still be useful if clinicians are trying to predict outcomes of patients in rehabilitation, but they are less useful for predicting outcomes in the early stages after stroke.

If a model is to be applicable in clinical practice, the predictors included in the model must be reproducible and easily obtained or available at the time the model is to be used [7,56]. Therapy intensity and length of stay have been shown to predict recovery of walking [24,28,66,84]. However, it is not possible to obtain data on therapy intensity or length of stay early after stroke. Therapy intensity may be interpreted differently in various stroke units, especially in different countries [24,28,66] which may cause the models to be poorly calibrated when applied internationally. In some prognostic models of upper limb function, predictors such as active range of motion and muscle strength of the upper limbs have been identified [72,81,82,85]. However these predictors were measured with 3D tracking systems or dynamometers which might not be available in routine clinical practice. Scales such as the Orpington Prognostic Score and the Orgogozo Score predict recovery in walking and upper limb function [44,77] but these scales are less commonly used than scales such as the National Institutes of Health Stroke Scale (NIHSS) [86] and so they may not be immediately available for making prognoses in many clinical settings.

Few studies that have developed prediction models of walking and arm recovery have used contemporary, robust variable selection processes. Most of these studies have either not conducted multivariate analyses [26,29,33,40,46,51,68,77,78,79] or have not provided a clear description of the variable selection process used to build their multivariate models [17,22,70,80]. In many studies where the variable selection process is clearly described, there is a large ratio of predictors to outcomes (e.g., there may be <10 events/predictor) [41,16], continuous predictors are dichotomised [14,24,41,76], or simplistic selection procedures (e.g., forward selection techniques) are used without accounting for the number of models considered in the selection process [16,24,41,45,76]. As a result, many models of walking and arm recovery prediction models may be susceptible to overfitting and predictor selection bias. It is likely that some of the predictors in these models will have spurious associations with the outcomes and that, when these models are used on new cohorts of patients they will produce inaccurate predictions [56,64,65]. Ultimately, the strongest protection against these problems comes from validation of the models prior to their use in clinical practice.

### 4.2. Arm Recovery: Critical Appraisal of Validation Studies

To date, none of the prediction models for walking has been validated. However one model for arm recovery—the proportional recovery model—has been validated [47]. This model was developed on 41 patients with a stroke and arm deficits (i.e., patients with Fugl Meyer-Upper Extremity (FM-UE) scores <66). Patients were recruited from a single inpatient stroke service in the USA. The model has subsequently been tested in two other cohorts in the USA and the Netherlands [87,88]. The outcome predicted in the proportional recovery model is the change in FM-UE scores (i.e., final FM-UE measured approximately 3 or 6 months after stroke minus initial FM-UE measured 24–72 h after stroke). The predictors included initial FM-UE, subcortical lesion volume, age and time to reassessment, though the final model only required input of initial FM-UE scores as the mean values of subcortical lesion volume, age and time to reassessment of the development sample was used in the final model in the development [47] and validation studies [87,88]. We have two concerns about the validation of the proportional recovery model. One is that the predictive accuracy of the proportional recovery model was inflated by excluding outliers, both in the development and validation stages. The development study reported high predictive accuracy (*r*^2^ = 0.90) after excluding 7 of 41 with severe initial impairment on the grounds that these were ‘outliers’ [47]. With the outliers included, the original adjusted *r*^2^ was only 0.47 [47]. The same issue arose in the validation studies, which reported high predictive accuracy after excluding 65 of 211 patients [87] and 7 of 30 patients [88]. Another issue is that, in one of the validation studies, the validation cohort included some participants who had been in the development cohort (14 of the 30 participants) [88]. The authors have consistently claimed that the proportional recovery model can accurately predict arm recovery in people who have had mild or moderate strokes, but not severe strokes. That claim would be stronger if the decision to apply the model only to people with mild or moderate strokes was explicitly made prior to the validation, and if the validation study was conducted on a completely independent sample to the development sample.

It appears that the proportional recovery model predicts outcomes well in some patients and not in others. Further analyses in the validation studies have revealed that patients with more severe stroke deficits (e.g., severe arm impairment [88], absent finger extension, facial palsy, severe leg impairment and total or partial anterior cerebral infarction [87]) had recovery profiles that did not fit the proportional recovery model. In a third study (not a validation study per se), Byblow et al. investigated the prognostic value of motor evoked potentials when used with the proportional recovery model [73]. The authors found the proportional recovery model had excellent predictive accuracy (*r*^2^ = 0.88–0.95) in 82 patients with present motor evoked potentials [73]. However, there were no head-to-head comparisons of the predictive accuracy of the model with and without neurophysiological measures at 6, 12 and 26 weeks and only a small number of patients with absent motor evoked potentials (*n* = 11) were included in the study. Further validation of the proportional recovery model with motor evoked potentials is required in a larger and more heterogeneous sample. Nonetheless, these results suggest that the inclusion of additional clinical or neurophysiological predictors might improve performance of the proportional recovery model in all patients with stroke.

### 4.3. Walking and Arm Recovery: Models That Might Be Ready for External Validation

In this section, we identify five well-developed models that might be ready for external validation.

These models (two on walking [20,25] and three on arm recovery [25,42]) were developed in the Netherlands [20,42] and Australia [25]. For the purpose of this review, we consider models which have recruited representative cohorts early after stroke, used clearly defined and standardised predictors in their models, and/or used robust methods to select predictors. The models we consider used standardised outcome measures and have high rates of follow-up, study features that suggest the findings are at low risk of bias [8,9,55,56]. All of these models reported the predictive accuracy of their models, which facilitates external validation.

#### 4.3.1. Walking

In the study by Veerbeek and colleagues [20], a sample of 154 non-ambulatory patients were recruited from nine hospital stroke units in the Netherlands. The predicted outcome was walking measured with the Functional Ambulation Category. The Functional Ambulation Category has been shown to be a reliable and valid tool for measuring walking ability in people who have had a stroke [89,90,91]. The study had a high follow-up rate (92% of survivors at 6 months). The authors found that early presence of lower limb muscle strength and sitting ability predicted recovery of walking six months after stroke [20]. These predictors were measured with the Motricity Index for leg (MI leg) and the Trunk Control Test item 3: sitting balance (TCT-s). These assessment tools are widely used in Europe and elsewhere [14,41,69,71] so it is possible to externally validate the model and it would ultimately be possible to use the model in these regions. The prediction models are as follows:

Model used <72 h:

Probability of walking at 6 months = 1/(1 + e^−0.982 + 2.691 TCT-s + 2.083 MI leg^)
(1)


Model used on day 5:

Probability of walking at 6 months = 1/(1 + e^−1.236 + 2.815 TCT-s + 1.609 MI leg^)
(2)


Model used on day 9:

Probability of walking at 6 months = 1/(1 + e^−2.226 + 3.629 TCT-s + 1.854 MI leg^)
(3)


Note that TCT-s and MI leg were dichotomised at 25. Therefore, TCT-s and MI leg have values of 0 (if the score is ≤25) or 1 (score >25). The sensitivity of the models ranged from 0.93 (95% CI = 0.86–0.96) to 0.94 (95% CI = 0.87–0.97) and specificity ranged from 0.63 (95% CI = 0.43–0.78) to 0.83 (95% CI = 0.64–0.93) [20].

In the second study on walking, by Kwah et al. [25] 114 non-ambulatory patients were recruited. Participants were sampled only from a single hospital in Australia. Sampling of consecutive patients reduced the risk of bias. It was found that age and severity of stroke (measured with NIHSS) predicted independent walking six months after stroke [25]. Age and NIHSS data are relatively easy data to obtain. If NIHSS scores are not available they can also be substituted with Canadian Neurological Scale (CNS) scores using the following regression equation [92]: *NIHSS = 23* − *2* × *CNS*. The risk of bias in this study was low because of the recruitment of a representative inception cohort, use of a standardised outcome measure (Motor Assessment Scale item 5 for walking) [93,94], high follow-up rates (95% of survivors at 6 months), consideration of only a small number of potential predictors, and the use of bootstrapping to shrink parameter estimates. The prediction model is:

Model used within 4 weeks of stroke:

Probability of walking at 6 months = 1/(1 + e^−11.02852 − 0.1053 age − 0.2436 NIHSS^)
(4)


The model had good discrimination (AUC of 0.84, 95% CI 0.77 to 0.92). Calibration was assessed using the Hosmer-Lemeshow test and showed no statistical significance for the model (0.70), indicating that there was no evidence of a failure of fit.

#### 4.3.2. Arm Recovery

The methods used in the study by Nijland and colleagues [42] were similar to methods used in the study by Veerbeek and colleagues [20] in that a representative acute stroke cohort (188 patients with hemiparesis) was recruited and predictors were standardised and easily collected in stroke units. The study also used a standardised outcome measure (the Action Research Arm Test) [95,96,97] to measure arm recovery and had high follow-up rates (93% of survivors). It was found that early presence of finger extension and shoulder abduction predicted arm recovery six months after stroke [42]. Predictors included the Fugl-Meyer finger extension (FM-FE) and the Motricity Index shoulder abduction (MI-SA). The multivariate prediction models are as follows:

Model used on day 2:

Probability of arm recovery at 6 months = 1/(1 + e^−1.119 + 2.807 FM-FE + 2.149 MI-SA^)
(5)


Model used on day 5:

Probability of arm recovery at 6 months = 1/(1 + e^−1.874 + 3.070 FM-FE + 3.075 MI-SA^)
(6)


Model used on day 9:

Probability of arm recovery at 6 months = 1/(1 + e^−1.815 + 3.224 FM-FE + 2.449 MI-SA^)
(7)


Note that FM-FE was dichotomised at 1 and MI-SA was dichotomised at 9. Therefore, if a patient has scores of FM-FE of less than 1 and MI-SA of less than 9, one inputs “0” for these variables. The sensitivity of the models ranged from 0.89 (95% CI = 0.85–0.92) to 0.95 (95% CI = 0.91–0.97) and specificity ranged from 0.83 (95% CI = 0.72–0.90) to 0.83 (95% CI = 0.74–0.89) [42].

The last two models of arm recovery were developed by Kwah et al. [25]. The outcomes were measured using items 7 and items 8 of the Motor Assessment Scale [36,93,94]. One model predicted the probability of being able to move a cup across a table six months after stroke (*N* = 65 patients) and the other model predicted the probability of being able to feed with a spoonful of liquid using the affected arm (*N* = 69 patients). The prediction models are:

Model used within 4 weeks of stroke:

Probability of moving a cup across the table at 6 months = 1/(1 + e^−4.8167 − 0.0533 age − 0.1240 NIHSS^)
(8)

Probability of feeding oneself with spoonful of liquid at 6 months = 1/(1 + e^−2.0855 − 0.2262 NIHSS^)
(9)


The AUCs for the prediction models were 0.73 (95% CI 0.59 to 0.87) for moving a cup and 0.82 (95% CI 0.70 to 0.94) for feeding. The Hosmer-Lemeshow test was not statistically significant for any model (0.74 for moving a cup, 0.38 for feeding oneself), so there was no evidence of a failure of fit.

It is important to note that these well-developed models are not without methodological limitations.

For example, in the models developed in the Netherlands, cohorts did not appear to be recruited consecutively. There is also a risk of overfitting or predictor selection bias in these models because continuous predictors were dichotomised and the number of predictors entered into the model exceeded the number of outcome events (thereby not fulfilling the widely used criterion of 10 outcome events per candidate predictor [60,61]). In the models developed in Australia, the cohorts were recruited from a single site which reduces the generalisability of the results. The sample sizes used in the arm recovery cohorts were also small (*n* < 70) which precluded the consideration of many predictors. The Australian studies used bootstrapping techniques to ‘shrink’ their models, but the bootstrapped models had poor calibration so subsequently only simplistic backwards selection models were used. Ultimately, these models must be externally validated before they can be used in clinical practice.

## 5. Arm Recovery: Do Neurophysiological and Neuroimaging Data Provide Additional Predictive Value over Clinical Data?

Neurophysiological and neuroimaging variables have been used to predict arm recovery. The most widely used neurophysiological predictor is the presence of motor evoked potentials (MEPs). MEPs are obtained using transcranial magnetic stimulation (TMS) which is a non-invasive form of brain stimulation. TMS of the primary motor cortex on the same side as the lesion activates muscles on the contralateral limb via the corticospinal tract, and the resulting MEPs are measured with surface electromyography [12]. Neuroimaging predictors include functional MRI and diffusion weighted MRI [5,12]. A common approach is to measure fractional anisotropy (FA) in specific sites in the brain, often the posterior limb of the internal capsules, using diffusion weighted MRI [49,73,98,99,100,101]. FA measures the directional preference of diffusion in white matter. Disruption of the corticospinal tracts decreases FA [98]. FA of the lesioned and non-lesioned hemispheres are often measured and presented as a ratio to predict arm recovery. This ratio is termed the FA asymmetry index. FA asymmetry index values of <0.15 and <0.25 have been used to predict arm recovery [49,73,98]. Recent studies that have used neuroimaging predictors to predict arm recovery have also measured “early fiber number ratio” [102] and “weighted corticospinal lesion load” [103,104] as markers of corticospinal tract integrity.

Two systematic reviews have been conducted on the prediction of arm recovery [3,6] and several more systematic reviews have investigated the prognostic value of neurophysiological and neuroimaging predictors on arm recovery [105,106,107]. The reviews conclude there is sufficient evidence to suggest neurophysiological and neuroimaging data have the potential to predict arm recovery [3,6,105,106,107]. However it remains debatable whether neurophysiological and neuroimaging data provide sufficient additional predictive information, compared to clinical data alone, to justify their use for prediction of arm recovery in clinical practice. This consideration is particularly important because there are already well-developed models that can predict arm recovery accurately using low-cost, readily available and easily collected clinical data [25,42].

A premise of studies that use neurophysiology and neuroimaging data to predict arm recovery is that clinical data alone do not provide adequate predictions of arm recovery, particularly in patients with initial severe arm impairment [73,103,108,109]. When this claim is made explicitly it is often supported by reference to the proportional recovery model or poorly conducted prognostic studies, rather than models which have shown good predictive accuracy [25,42]. In addition, comparisons of predictive accuracy of models based on clinical data and models based on neurophysiological or neuroimaging data consistently demonstrate that prediction models based on clinical data have similar or higher predictive accuracy in predicting arm recovery [75,99,103,104,110]. When neurophysiological or neuroimaging data are added to models containing clinical data the increase in predictive accuracy is small [75,99,110] (e.g., increments in *r*^2^ of 5%–9% [75,99])—probably too small to justify the expense and inconvenience of these investigations. (Note that in the study by Byblow et al., no comparisons of predictive accuracy were made between the proportional recovery model and the proportional recovery model with motor evoked potentials. So the additional predictive value of motor evoked potentials in predicting arm recovery in *all* patients has not yet been unambiguously established [73].)

A notable prediction model is the PREP algorithm [49]. This model uses clinical predictors that are similar to those identified by Niljand et al. [42] (muscle strength of shoulder abduction and finger extension) as well as neurophysiological (presence of MEPs) and neuroimaging (FA asymmetry index <0.15) predictors. The algorithm involves a step-by-step approach. First the strength of the shoulder abductor and finger extensor muscles is measured using the Medical Research Council muscle grades and, if the combined score is less than 8, TMS is used. If MEPs are absent, MRI is used. The PREP model had a positive predictive value of 88% and a negative predictive value of 83% (specificity 88% and sensitivity 73%; positive likelihood ratio 6.1. negative likelihood ratio 0.31) on the development set for complete recovery at 3 months. This is a potentially useful level of predictive accuracy, particularly also because the model predicted a high level of arm recovery (ARAT ≥ 50/56) compared to that in Niljand et al. (ARAT ≥ 10/56) [42]. However, the predictive accuracy of the models at each step of the algorithm was not reported so it is not clear that the increase in predictive accuracy provided by TMS and DTI is sufficient to make the acquisition of these time- and labour-intensive measurements worthwhile. It is also worth noting that the sensitivity and specificity of the PREP model in predicting complete recovery were actually lower than the sensitivity and specificity of the model developed by Niljand et al. [42], which brings into question whether neurophysiological and neuroimaging data are necessary for accurate predictions of arm recovery.

A potential advantage of basing predictions on neurophysiology and neuroimaging data rather than clinical data is that it is possible to collect neurophysiology and neuroimaging data from patients with cognitive or neuropsychological deficits [12,75,110]. In these patients it may be hard to conduct clinical assessments that involve self-reporting or performance of complex tasks. However, studies that have predicted arm recovery with neurophysiological and neuroimaging data have often excluded patients with cognitive deficits [49,73,98,99,100,101,103,104,109]. Moreover, patients with previous strokes or contraindications to TMS or MRI are also often excluded from TMS and MRI studies [49,73,75,98,99,100,101,102,103,109,110]. Consequently it is not yet clear that neurophysiological and neuroimaging data can aid prognosis in cognitively or neuropsychologically impaired patients.

## 6. Conclusions and Recommendations for Future Studies

If a prediction model is to be used in clinical practice, it must be well developed and externally validated prior to use in clinical practice. While many models predicting death and severe disability after stroke have been well developed and externally validated [111,112,113,114], few models for predicting walking and arm recovery have been well developed. One model for arm recovery (the proportional recovery model) has been externally validated though results from development and validation studies suggest that the model does not predict well in *all* patients with stroke. This model does appear to predict outcomes well in people with less severe strokes. In this review we have identified five models [20,25,42] that are potentially ready for external validation.

It will benefit future researchers in prognostic research to have access to large datasets from stroke registries such as the Australian Stroke Clinical Registry [115]. Stroke registries can provide representative cohorts and large data sets that permit rigorous external validation of models. Regarding the choice of predictors, future researchers should consider the National Institute of Neurological Disorders and Stroke (NINDS) Stroke Common Data Element (CDE) project that recommends a common set of data to collect in studies involving people with stroke and transient ischemic attacks [116]. That would ensure data on predictors are standardised and more easily obtained in routine clinical practice, and it would help to facilitate external validation of models in other stroke cohorts.

Although there is no single recommended approach to building a multivariate prediction model, several techniques are known to lead to overfitting and predictor selection bias. Researchers conducting prognostic research should use contemporary techniques and refer to guidelines on the reporting and conduct of prediction modelling studies [8,9,56] in order to develop robust prediction models. A good example of a rigorously developed and validated prediction model is the iScore model used to predict mortality after stroke112]. The iScore model has been externally validated and has an acceptable predictive accuracy (c-statistics ≥0.78) [111,117]. It can be implemented with a web application [118] or a smartphone application [119]. Distribution of software for calculating prognoses using web sites or smartphones will facilitate use of prediction models in practice.

Neurophysiology and neuroimaging data have an important role to play in diagnosis and in understanding mechanisms of arm recovery [120,121,122]. While many studies suggest neurophysiology and neuroimaging data can be helpful in predicting arm recovery [3,6,105,106,107], there is no strong evidence that models containing neurophysiology or neuroimaging data predict arm recovery *better* than models containing clinical data, or that the addition of neurophysiology or neuroimaging data to models containing clinical predictors provide sufficient increase in predictive accuracy to warrant their use. Researchers who develop prediction models based on neurophysiological or neuroimaging data should routinely investigate whether the neurophysiological or neuroimaging data provide better predictive accuracy than models based on clinical data alone. Exemplary studies of prediction models for stroke include the neuroimaging studies conducted by Johnston et al. [123,124,125] and Schiemanck et al. [126]. In their studies predicting outcomes [123,124,125] and independence in activities of daily living [126] after stroke, neuroimaging data (i.e., CT or MRI scans of infarct volume) either did not improve the predictive accuracy of the model with clinical predictors [123,124,126] or produced only small and unimportant increases in predictive accuracy [125].

Readers should be aware that we have only provided a brief critical appraisal of development studies in walking and arm recovery. Other factors such as outcome measurement, sample size, missing data, blinding of predictor measurement, timing of predictor measurement and handling of predictors in the model can also influence the risk of bias and/or applicability of a prediction model [56].

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
