# Peer review of "Prediction of Walking and Arm Recovery after Stroke: A Critical Review"

_brainsci, 2016, doi:10.3390/brainsci6040053_

Round 1

Reviewer 1 Report

This is a generally well-written and timely review. The authors have covered most of the relevant literature, and make important points concerning the development and validation of prediction models. However, the discussion and critique seems to be selective in places, and this needs to be addressed to provide a more balanced review. Further comments below.

The authors ought to provide their search strategy for this review. It would also be helpful to clearly state whether the predictors identified by each of the studies reviewed were predicting future function at the acute or sub-acute stage, or were reporting relationships at a single point in time.

The Introduction makes the point that it would be helpful if clinicians could make accurate recovery predictions for patients. However, the authors don’t comment on the fact that the majority of the models they review identify predictors based on data from groups of patients, and are unable to make predictions for individual patients. How is a linear or logistic regression equation useful to the clinician? What should a therapist do with the equations given on Line 193?

Line 87: If we only investigate and include predictors that are already easily accessible and available at the time the model is intended to be used, this limits the development of new predictors. With new technologies becoming more widely available, it is unhelpful to discourage people from thinking about predictors that are not currently easily accessible and available. Why limit the future by the present?

Line 119: Selecting patients from those referred for rehabilitation is not a problem if the prediction is the response to rehabilitation. Clarity around what is being predicted, and why, is needed here.

Section 2b: The criticism of the proportional recovery model is missing more recent studies. Subsequent work cited later (Byblow et al. 2015) has shown that adding a predictor (MEP+/-) clearly distinguishes between patients whose spontaneous recovery is proportional (and 70%), and those whose recovery is not. This is an important improvement to the model that counters the criticisms in this section.

Line 291: A small increase in the fit of the model (r2) is not the only one way to evaluate the usefulness of a model. The models the authors are referring to here are trying to explain variance in a group of patients rather than make an accurate prediction for an individual patient. As above, explaining a little more variance in a group of patients isn’t particularly useful for a clinician. Being able to detect whether a severely impaired patient will make some, or no, recovery of motor ability is far more relevant – and this question can’t be answered by clinical measures alone. The authors note in section 2b that the proportional recovery model is one of the stronger models currently available, but ignore that neurophysiology and neuroimaging measures have made a critical contribution to this model, allowing predictions for more severely impaired patients that cannot be made by clinical measures alone. This completely contradicts their argument.

The authors have provided a useful critique of several aspects of studies that predict motor recovery after stroke, but they have failed to critique WHAT is being predicted. Are the measures, states or outcomes being predicted actually useful and meaningful for individual patients? This is important throughout, and particularly relevant around Line 308, where the authors compare the predictive accuracy of two upper limb models. Is being more accurate at predicting a more coarse outcome an advantage?

Author Response

REVIEWER 1

Comments and Suggestions for Authors

This is a generally well-written and timely review. The authors have covered most of the relevant literature, and make important points concerning the development and validation of prediction models. However, the discussion and critique seems to be selective in places, and this needs to be addressed to provide a more balanced review. Further comments below.

Comment: The authors ought to provide their search strategy for this review.

Response: We have adopted this suggestion. The new text (in Appendix on pg. 9-10, lines 438-443) reads as below:

Appendix (Search strategy): References for this review were identified by searching PubMed from January 1970 to July 2016. The search syntax included (predict* OR prognos* OR probability) AND ((walk OR gait OR ambulat* OR mobility) OR (arm OR upper limb)) AND (stroke). Additional references were identified by inspecting reference lists of relevant articles and authors’ personal libraries. Studies were included if they were cohort studies, measured outcomes of walking and arm recovery and were published in English.

If the placement of text in an Appendix does not align with journal requirements, we are happy for it to be placed elsewhere in the review.

Comment: It would also be helpful to clearly state whether the predictors identified by each of the studies reviewed were predicting future function at the acute or sub-acute stage, or were reporting relationships at a single point in time.

Response: We have clarified in the inclusion criteria that only cohort studies were included (see previous response on search strategy). Cross-sectional studies (i.e., studies that collected data at a single time point) cannot offer insights into prediction and hence are not included in this review.

In order to keep the review brief, we have not reported timing of predictor measurement for every study. However we have reported timing of predictor measurement for the models that are well developed. We agree that timing of predictors is an important factor to consider when critically appraising a prediction model and have now made this point explicitly. The new text (on pg. 9, lines 432-436) reads as below:

Readers should be aware that we have only provided a brief critical appraisal of development studies in walking and arm recovery. Other factors such as outcome measurement, sample size, missing data, blinding of predictor measurement, timing of predictor measurement and handling of predictors in the model can also influence the risk of bias and/or applicability of a prediction model56.

Comment: The Introduction makes the point that it would be helpful if clinicians could make accurate recovery predictions for patients. However, the authors don’t comment on the fact that the majority of the models they review identify predictors based on data from groups of patients, and are unable to make predictions for individual patients.

Response: All of the prediction models considered here are models of the outcomes of individual patients.

Comment: How is a linear or logistic regression equation useful to the clinician? What should a therapist do with the equations given on Line 193?

Response: A linear or logistic regression equation provides information about a patient’s prognosis at a later time point. In context of our review, a linear regression equation provides clinicians with patient-specific predictions of Fugl Meyer-Upper Extremity outcomes while a logistic regression equation provides clinicians with patient-specific predictions of the probability of walking and arm recovery.

As outlined in the first paragraph of the Introduction, accurate predictions of walking and arm recovery (at a later time point) can help clinicians make decisions on goal-setting, therapy selection and discharge planning in the early stages after stroke.

Comment: Line 87: If we only investigate and include predictors that are already easily accessible and available at the time the model is intended to be used, this limits the development of new predictors. With new technologies becoming more widely available, it is unhelpful to discourage people from thinking about predictors that are not currently easily accessible and available. Why limit the future by the present?

Response: We have no objection to the use of new predictors/technologies as long as they are, or might be expected to become, easily accessible/available and add to the predictive accuracy of current well-developed models. These opinions are also held by experts in prognostic research1,2 and stroke research3-6.

Comment: Line 119: Selecting patients from those referred for rehabilitation is not a problem if the prediction is the response to rehabilitation. Clarity around what is being predicted, and why, is needed here.

Response: We agree with the reviewer that there is some value in models developed on rehabilitation cohorts. The new text (on pg. 4, lines 166-168) reads as below:

Models developed on trial cohorts or rehabilitation cohorts might still be useful if clinicians are trying to predict outcomes of patients in rehabilitation, but they are less useful for predicting outcomes in the early stages after stroke. 

Comment: Section 2b: The criticism of the proportional recovery model is missing more recent studies. Subsequent work cited later (Byblow et al. 2015) has shown that adding a predictor (MEP+/-) clearly distinguishes between patients whose spontaneous recovery is proportional (and 70%), and those whose recovery is not. This is an important improvement to the model that counters the criticisms in this section.

Response: We thank the reviewer for highlighting the study by Byblow et al. We have incorporated their results into the review. The new text (on pg. 5, lines 220-234) reads as below:

It appears that the proportional recovery model predicts outcomes well in some patients and not in others. Further analyses in the validation studies have revealed that patients with more severe stroke deficits (e.g. severe arm impairment88, absent finger extension, facial palsy, severe leg impairment and total or partial anterior cerebral infarction89) had recovery profiles that did not fit the proportional recovery model. In a third study (not a validation study per se), Byblow et al investigated the prognostic value of motor evoked potentials when used with the proportional recovery model74. The authors found the proportional recovery model had excellent predictive accuracy (r2 = 0.88-0.95) in 82 patients with present motor evoked potentials74.However, there were no head-to-head comparisons of the predictive accuracy of the model with and without neurophysiological measures at 6, 12 and 26 weeks and only a small number of patients with absent motor evoked potentials (n = 11) were included in the study. Further validation of the proportional recovery model with motor evoked potentials is required in a larger and more heterogeneous sample. Nonetheless, these results suggest that the inclusion of additional clinical or neurophysiological predictors might improve performance of the proportional recovery model in all patients with stroke.

The new text (on pg. 8, lines 356-360) reads as below:

(Note that in the study by Byblow et al, no comparisons of predictive accuracy were made between the proportional recovery model and the proportional recovery model with motor evoked potentials. So the additional predictive value of motor evoked potentials in predicting arm recovery in all patients has not yet been unambiguously established74.)

Comment: Line 291: A small increase in the fit of the model (r2) is not the only one way to evaluate the usefulness of a model. The models the authors are referring to here are trying to explain variance in a group of patients rather than make an accurate prediction for an individual patient. As above, explaining a little more variance in a group of patients isn’t particularly useful for a clinician. Being able to detect whether a severely impaired patient will make some, or no, recovery of motor ability is far more relevant – and this question can’t be answered by clinical measures alone.

Response: The r2 statistic (the proportion of the variance explained by the model) quantifies the predictive accuracy of the model7,8. Thus “explaining a little more variance” means increasing the accuracy of predictions. That is both relevant and useful for clinicians.

Comment: The authors note in section 2b that the proportional recovery model is one of the stronger models currently available, but ignore that neurophysiology and neuroimaging measures have made a critical contribution to this model, allowing predictions for more severely impaired patients that cannot be made by clinical measures alone. This completely contradicts their argument.

Response: We thank the reviewer for highlighting the study by Byblow et al. Please see earlier response on pg.3.

Comment: The authors have provided a useful critique of several aspects of studies that predict motor recovery after stroke, but they have failed to critique WHAT is being predicted. Are the measures, states or outcomes being predicted actually useful and meaningful for individual patients? This is important throughout, and particularly relevant around Line 308, where the authors compare the predictive accuracy of two upper limb models. Is being more accurate at predicting a more coarse outcome an advantage?

Response: We acknowledge the importance of defining outcomes. We added two new paragraphs. In these paragraphs, we have also addressed the difficulty of defining arm recovery. The new text (on pg. 2-3, lines 68-109) reads as below:  

1. Definitions and measurements of walking and arm recovery

Most studies investigating recovery of walking after stroke have focused on the ability to walk independently13-22. The operationalization of this construct varies slightly with regard to the distance covered, the type of aids needed and the type of surfaces. For example, independent walking has been defined as “independence of gait and stair climbing with or without the use of some form of assistive device”16, “walks 5m without any assistance or walks with aid”22 and “imperfect gait, but could be independently ambulant, usually with a walking aid”15. Various measurement tools have been used to measure walking in stroke. Some authors have used walking-specific measurement tools like the Functional Ambulation Category14, 16, 17, 20, 23, the Rivermead Mobility Index19, 24 or the walking items of scales that measure stroke severity (e.g. Scandinavian Stroke Scale)22, motor function (e.g. Motor Assessment Scale)25 or activities of daily living (e.g. the Barthel Index or the Functional Independence Measure)21, 26-33. Despite the heterogeneity of these measurement tools, it is easy to extract data on walking from them. Independent walking is equivalent to a score of ≥ 4/5 on the Functional Ambulation Category14, 16, 17, 20, 23, a score of ≥ 3/6 on item 5 of the Motor Assessment Scale25, a score of ≥ 9/12 on the gait item of Scandinavian Stroke Scale22 and a score of 15/15 on the mobility item of the Barthel Index28, 29. Studies that measured walking outcomes were included in this review.

For arm recovery, there is less consensus on an operational definition. It is generally agreed that measures of arm recovery should reflect a patient’s ability to use his or her arm in everyday life6, 34-36. This is certainly considered an important goal to patients after stroke37. It is difficult, however, to define arm recovery due to the wide range of actions or tasks associated with functional use of the arm. Therefore the term arm recovery in this review will be used broadly to include measures of body functions or structure (e.g. muscle strength) or activities (e.g. ability to carry a tray of food across a crowded room)6, 38. Various measurement tools used in the literature to measure arm recovery include arm-specific measurement tools like the Action Research Arm Test39-43, the Nine Hole Peg Test43-46 and the upper limb items of scales that measure body functions or structures (e.g. Motricity Index45, Fugl-Meyer45, 47) or activities (e.g. Motor Assessment Scale25, 30, 48). In prognostic studies in stroke, arm recovery has been defined by cut-off scores or change scores on these scales. Examples include cut-off scores of 10/5641, 42, 35/5639, 50/5649 on the Action Research Arm Test, 5/6 on upper limb items 7 and 8 of Motor Assessment Scale25, 18/18 on upper limb items 6 to 8 of Motor Assessment Scale48 and change scores of upper limb items of Motricity Index45 and Fugl-Meyer45, 47. The problem with attributing a score to the definition of arm recovery is that readers have to be familiar with the scale to know what the score represents. Some have argued that a low score (e.g. 10/56 on the Action Research Arm Test) may not be reflective of meaningful arm function5, 39. Some authors have also used activities of daily living scales (e.g. the feeding, dressing and toileting items of the Barthel Index) to measure arm recovery50, 51. These scales are, however, insensitive to recovery of function in the hemiparetic arm since they allow use of the intact arm6, 34, 52. For this reason, tests that allow use of both arms (e.g. the Wolf Motor Function Test, the Frenchay Arm Test, the Arm Motor Ability Test and the Bell Test) may be less preferred as measures of arm recovery52. Studies that measured outcomes of hemiparetic arm recovery were included in this review. Studies that used activities of daily living scales or tests that allow use of both arms were not included in this review.

For the statement in Line 308 in the previous version of the manuscript (now pg. 8, lines 370-371), we have also added new text:

This is a potentially useful level of predictive accuracy, particularly also because the model predicted a high level of arm recovery (ARAT ≥ 50/56) compared to that in Niljand et al (ARAT ≥ 10/56)42.

In response to the reviewer’s suggestions, minor editorial changes have also been made to the manuscript. These are all recorded as tracked changes in the current Word document. Sections have been re-ordered. Minor editorial changes can be found on

·       Pg.1, lines 19-20,

·       pg.2, lines 57-64,

·       pg.5, lines 203-206, and

·       pg. 9, lines 389-395.

Reviewer 2 Report

In this review, the authors start off with a discussion of points to consider in development of a prediction model, including the recruitment of representative cohorts, use of appropriate predictors, and the selection of predictors to include in the model, and discuss key elements in model validation. They undertake a critical review of existing models including the proportional recovery model and the existence of shoulder abduction and finger extension (SAFE) for arm recovery, and the incorporation of neurophysiological and neuroimaging data into models of recovery. In their appraisal, they note that most of the models that have been developed have methodological limitations in the way that patients were included or in the selection of appropriate predictors. Only one model for arm recovery, proportional recovery, has been validated to date, and was developed and validated with exclusion of some“outliers” with severe stroke, thus is unable to predict recovery for all stroke patients. Other recovery models are discussed but have not yet been validated externally. Furthermore, while neurophysiological and neuroimaging data have been incorporated into models, the authors note that “it is not clear whether the increase in predictive accuracy provided by [these data] is sufficient to make the acquisition of these time- and labour-intensive measurements worthwhile”.

Major comments:

This is a clear, well-written overview of predictive models for motor recovery after stroke. I appreciate the discussion of methodological considerations in the development and validation of these models and the critiques of existing models in this context.

A discussion of particular challenges in developing models of post-stroke recovery would be helpful for readers. As seen from the one validated model of proportional model, there may be subgroups of patients that follow different recovery courses, and recovery from stroke is complex; should we expect to find a unifying model for all patients?  Furthermore, should therapy duration or intensity be considered in the development of prediction models, if it is generally believed that therapy impacts recovery in some way? 

The authors state that accurate predictions can aid is goal setting and selection of therapies, and give an example that the clinicians may opt not to intensively train gait if the prediction is that a given patient would be unable to walk independently at 6 months. While models would be helpful in this regard, we should also use some caution when using models that do not account for rehabilitative interventions to predict response to these interventions.

Author Response

REVIEWER 2

Comments and Suggestions for Authors

In this review, the authors start off with a discussion of points to consider in development of a prediction model, including the recruitment of representative cohorts, use of appropriate predictors, and the selection of predictors to include in the model, and discuss key elements in model validation. They undertake a critical review of existing models including the proportional recovery model and the existence of shoulder abduction and finger extension (SAFE) for arm recovery, and the incorporation of neurophysiological and neuroimaging data into models of recovery. In their appraisal, they note that most of the models that have been developed have methodological limitations in the way that patients were included or in the selection of appropriate predictors. Only one model for arm recovery, proportional recovery, has been validated to date, and was developed and validated with exclusion of some“outliers” with severe stroke, thus is unable to predict recovery for all stroke patients. Other recovery models are discussed but have not yet been validated externally. Furthermore, while neurophysiological and neuroimaging data have been incorporated into models, the authors note that “it is not clear whether the increase in predictive accuracy provided by [these data] is sufficient to make the acquisition of these time- and labour-intensive measurements worthwhile”.

Major comments:

This is a clear, well-written overview of predictive models for motor recovery after stroke. I appreciate the discussion of methodological considerations in the development and validation of these models and the critiques of existing models in this context.

Comment: A discussion of particular challenges in developing models of post-stroke recovery would be helpful for readers. As seen from the one validated model of proportional model, there may be subgroups of patients that follow different recovery courses, and recovery from stroke is complex; should we expect to find a unifying model for all patients? 

Response: Based on the proportional recovery model, it does appear subgroups of patients follow different recovery courses. We now discuss this in some detail (see response to Reviewer 1’s comment on Section 2b).  

Comment: Furthermore, should therapy duration or intensity be considered in the development of prediction models, if it is generally believed that therapy impacts recovery in some way? The authors state that accurate predictions can aid is goal setting and selection of therapies, and give an example that the clinicians may opt not to intensively train gait if the prediction is that a given patient would be unable to walk independently at 6 months.

Response: We agree: there is evidence from randomized trials that therapy intensity impacts recovery. However as outlined in Section 2a 2nd para, therapy intensity is not an ideal predictor as it is not easily obtained or available early after stroke. It is also difficult to standardise the measurement of therapy intensity across stroke units in different countries.

Another justification for ignoring therapy intensity in prediction models is that the effects of therapy are typically small compared to the variability of outcomes. Systematic reviews of randomised trials suggest few if any therapies have standardized mean effect sizes as large as 0.59,10, yet a therapy with a standardized mean effect size of 0.5 explains only 25% of the variance of outcomes.

Comment: While models would be helpful in this regard, we should also use some caution when using models that do not account for rehabilitative interventions to predict response to these interventions.

Response: We agree. However the effects of specific interventions (e.g. constraint-induced movement therapy11-13) are beyond the scope of this review.

Round 2

Reviewer 1 Report

The authors have provided a thoughtful and detailed response to comments.However, the authors seem to be avoiding the question of how the clinician is supposed to use a linear or logistic regression equation. Is the therapist supposed to remember the equations provided around line 257 and pull out their phone to do some maths? The limited clinical usefulness of regression equations must be addressed.

Author Response

Please see our response attached in a Word document.
